# Investigating the relationship between thought interference, somatic passivity and outcomes in patients with psychosis: a natural language processing approach using a clinical records search platform in south London

Thibault Thierry Magrangeas [1], Anna Kolliakou [1], Jyoti Sanyal,[2] Rashmi Patel [1,2] Robert Stewart[1,2]

[1]Institute of Psychiatry Psychology and Neuroscience, King's College London, London, UK
[2]South London and Maudsley NHS Foundation Trust, London, UK

**Correspondence to**
Dr Thibault Thierry Magrangeas;
thibault.magrangeas@kcl.ac.uk

## ABSTRACT

**Objectives** We aimed to apply natural language processing algorithms in routine healthcare records to identify reported somatic passivity (external control of sensations, actions and impulses) and thought interference symptoms (thought broadcasting, insertion, withdrawal), first-rank symptoms traditionally central to diagnosing schizophrenia, and determine associations with prognosis by analysing routine outcomes.

**Design** Four algorithms were developed on deidentified mental healthcare data and applied to ascertain recorded symptoms over the 3 months following first presentation to a mental healthcare provider in a cohort of patients with a primary schizophreniform disorder (ICD-10 F20-F29) diagnosis.

**Setting and participants** From the electronic health records of a large secondary mental healthcare provider in south London, 9323 patients were ascertained from 2007 to the data extraction date (25 February 2020).

**Outcomes** The primary binary dependent variable for logistic regression analyses was any negative outcome (Mental Health Act section, >2 antipsychotics prescribed, >22 days spent in crisis care) over the subsequent 2 years.

**Results** Final adjusted models indicated significant associations of this composite outcome with baseline somatic passivity (prevalence 4.9%; adjusted OR 1.61, 95% CI 1.37 to 1.88), thought insertion (10.7%; 1.24, 95% CI 1.15 to 1.55) and thought withdrawal (4.9%; 1.36, 95% CI 1.10 to 1.69), but not independently with thought broadcast (10.3%; 1.05, 95% CI 0.91 to 1.22).

**Conclusions** Symptoms traditionally central to the diagnosis of schizophrenia, but under-represented in current diagnostic frameworks, were thus identified as important predictors of short-term to medium-term prognosis in schizophreniform disorders.

## INTRODUCTION

Schizophrenia was originally defined by Schneider around a core group of first-rank symptoms (FRS), which include

## STRENGTHS AND LIMITATIONS OF THIS STUDY

⇒ This study was carried out on a very large patient sample and identified symptoms from routine care instead of specifically recruited and therefore less representative cohorts, and our applications achieved high-performance metrics.

⇒ The patient sample was drawn from an inner urban area in South London, so results may not be entirely generalisable to the rest of the population.

⇒ The recording of exposure variables relied on symptoms being elicited and noted by clinicians in routine practice and therefore cannot be assumed to be equivalent to those ascertained in a more research-focused interview, just as covariates such as diagnosis also were derived from routine clinical records and should not be assumed to equate research diagnostic criteria; however, we feel that inaccuracies in exposure ascertainment were more likely to obscure findings of interest through non-differential measurement error rather than give rise to spurious associations.

⇒ Although multiple confounding factors were accounted for, our model-fit measures demonstrate model reliability but highlight residual unexplained variance possibly due to the influence of other psychotic symptoms as well as additional external factors that may affect outcomes and should be further investigated; our model is useful in correlating patient outcomes with symptoms at first presentation but it is not possible to infer stability of the clinical construct over time as the patient is only assessed once.

⇒ Outcomes analysed were restricted to those most readily available from the source data and most likely to be clinically informative; however, they cannot be viewed as exhaustive and there may be elements of prognosis that were not captured in this study, including interactions between outcomes and previous adverse events that may influence symptom occurrence and recording.

auditory hallucinations (commonly characterised as but not limited to hearing one or multiple voices talking to or about the patient), delusions of perception (such as the sensation of being externally controlled) or thought interference (the sensation of having thoughts inserted, withdrawn or externally broadcasted against the patient's wishes).[1] These symptoms were identified as central to diagnosing schizophrenia as they were observed in the majority of cases. Still today, recent studies have consistently found FRS in the form of thought interference and somatic passivity to be core symptoms of psychotic disorders in general and schizophrenia specifically.[2–4] A comparative study of 13 different diagnostic systems identified a triad of positive symptoms strongly correlated with a schizophrenia diagnosis across all scales, the presence of two of which was sufficient to diagnose 78% of patients; these were auditory hallucinations, disturbances of affect and 'passivity feelings' (defined as 'any delusion of being influenced or interfered with by imaginary forces from outside, whether somatically or mentally', and therefore, assumed to include thought interference)[5]; these symptoms were all already described in Bleuler's 1911 definition of schizophrenia,[6] demonstrating historical consistency. Somatic passivity and thought interference have also been reported to be positively correlated and predict the occurrence of schizophrenia or schizoaffective disorder.[7 8]

Despite this, ongoing research into the complex mechanisms underpinning psychosis manifestations has highlighted the wider range of symptoms that can occur and encouraged a new approach focused beyond these core FRS. For example, different symptoms have been observed to occur in clusters, such as somatic passivity and persecutory delusions[9] or auditory verbal hallucinations.[10] Environmental factors also seem to influence symptom occurrence, leading to debate around the cross-cultural relevance of FRS.[11] In this context, recent changes to the Diagnostic and Statistical Manual of Mental Disorders, Fifth Edition (DSM-5 including the removal of FRS from diagnostic criteria[12 13] have been welcomed as a valuable push towards a better understanding of individual symptoms; however, limitations remain as this new focus has been criticised for not fully addressing the relationship between symptoms; conflicting with most existing guidelines for early symptom detection in schizophrenia that still heavily rely on FRS; and diverting attention away from core psychopathological processes at the disorder's root.[14]

Importantly, schizophrenia symptoms can directly influence both behaviour and consequently patient outcomes; for example, indirectly leading to violence through the occurrence of delusions.[15] FRS absence has been found to correlate with better outcomes, leading to a proposed redefinition of schizophrenia as a continuum, exhibiting a positive linear relationship between psychotic symptoms and outcomes.[16–18] Furthermore, the prevalences of individual symptoms appear to vary between early-onset and late-onset cases, reflecting complex interactions between

symptoms and disease development.[19–21] This evidence suggests that patient outcomes are better in schizophrenia in the absence of FRS[17] but offers no insight into prognosis once present or into relationships between symptoms. While some authors have identified differing effects on outcomes, with passivity predicting a worsened prognosis in contrast to interference,[17] others have highlighted a possible link, as both phenomenologically reflect a belief in external control, suggesting they may have a synergistic effect on outcomes[8 22 23] or even constitute components of the same delusion[10 24] or syndrome.[25] Overall, these studies highlight the significance of positive symptoms individually and as a group, suggesting possible correlations and reflecting the overlap in their occurrence, although the degree of overlap is rarely quantified.

The evidence highlighted here reflects the importance of clarifying the role of FRS in prognosis to understand which aspects of schizophrenia should be at the core of its diagnosis. Modern technologies have allowed the development of new tools allowing the analysis of large datasets using automated software. This is the goal of the South London and Maudsley (SLaM) National Health Service (NHS) Foundation trust Clinical Record Interactive Search (CRIS) platform was developed in 2008 to enable research on a deidentified copy of the Trust's electronic health records (used across all services since 2006) within a robust governance and data security framework.[24] All the clinical notes of patients are recorded in a deidentified format on this database. Natural language processing (NLP) algorithms are then used to study these documents; these are algorithms that can be trained to recognise entities of interest and their context in free text, for example the mention of 'thought interference' in a way that indicates that a patient is experiencing this symptom.

This study aimed to develop and evaluate four NLP algorithms to detect the recording of thought interference subcomponents (broadcasting, insertion, withdrawal) and somatic passivity across this database. Other algorithms had already been developed to capture the recorded occurrence of other important positive symptoms of psychosis (paranoia, auditory hallucinations, persecutory delusions) and were used to define potential covariates. We focused on individuals diagnosed with a schizophreniform disorder and analysed patient outcomes in this population. Our focus on thought interference and somatic passivity as particular symptoms aimed to clarify the interactions described previously which suggested a particularly strong relationship between these symptoms; in addition, they were chosen as potentially tractable to extraction at scale from routine mental healthcare data via NLP. Being described in relatively consistent language in clinical records, they made a pragmatic choice for further investigation in contrast to other symptoms such as running commentary hallucinations which are typically described with greater linguistic heterogeneity (ie, based on the individual experience of patients rather than a set vocabulary), and therefore, harder to study with this approach.

## METHODS

### Context

We used data extracted from the electronic mental health clinical notes records of patients at the SLaM NHS Foundation Trust, one of the largest mental health service providers in Europe, serving a geographic catchment of four South London boroughs (Southwark, Lambeth, Lewisham, Croydon) with around 1.3 million residents. The Trust provides care to 37 500 active patients, with 12 000–14 000 clinical events per week covering a diverse range of interventions in the community and in the inpatient setting.[26 27] SLaM's Clinical Record Interactive Search (CRIS) platform was developed in 2008 to enable research on a deidentified copy of the Trust's electronic health records (used across all services since 2006) within a robust governance and data security framework.[24] CRIS has been described in detail previously[28 29] and has supported over 250 research publications to date.

### Patient sample

All patients aged 16–95 with a primary F20–F29 disorder diagnosis according to the 10th revision of the International Statistical Classification of Diseases and Related Health Problems (ICD-10) criteria recorded from 2007 to extraction date (25 February 2020) were included. This age range was chosen to cover patients transitioning to adult mental healthcare, commonly occurring from age 16; patients with no recorded date of birth appear on the system as aged 100+ and represented all the individuals above age 95; they were, therefore, excluded to avoid skew when correcting for age. According to these criteria, 9536 patients were identified. Thought interference and somatic passivity occurrence in 3 months following first presentation at SLaM (defined as first face-to-face contact) was sought. If multiple diagnoses were then recorded for the same patient, first diagnosis was used. This cut-off was chosen to allow time for clinicians to assess all patients and record their symptoms accurately and focus on patients presenting with these symptoms as first presentation rather than later development. Participants who died within this period were excluded. Of the initial 9536 patients, 9323 (97.8%) were included, 57.9% of whom were male. Mean age at inclusion was 39.9 years (SD=17.3).

### Exposure variables and software development

CRIS data have been enhanced over the last 10 years by a range of NLP algorithms allowing a hitherto unavailable level of detail to be generated at scale from routine healthcare data, taking advantage of the level of detail recorded in extensive text fields. The algorithms, over 90 to date, are described individually in an open-access online catalogue,[30] including detailed information on definitions and performance. A particular objective has been to provide, at scale, broader phenotypic information on people presenting with mental disorders than that supplied by clinician-assigned diagnostic codes, through the extraction of recorded symptoms with 60 algorithms

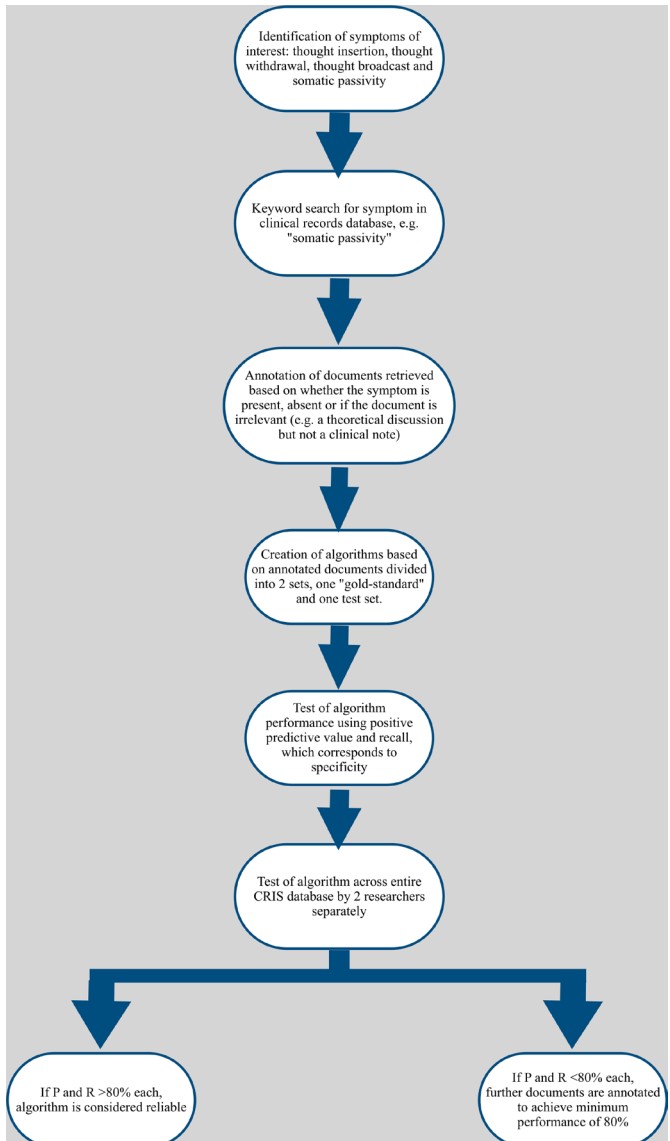

**Figure 1** Flow chart illustrating the development of algorithms via the TextHunter platform. TextHunter allows a training dataset of text strings containing wording of interest to be presented in an easy visual form for researcher annotation. The development process is identical for all algorithms relating to symptoms recorded in clinical text. CRIS, Clinical Record Interactive Search.

to date developed to ascertain individual symptoms across psychotic and affective disorders.[31] Contributing to this wider initiative, and with the specific rationale in mind for this study, we developed four new algorithms to identify instances of recorded somatic passivity and thought insertion, withdrawal and broadcasting in CRIS using the TextHunter platform,[32] a tool used to present short segments of text to an annotator in order to provide rapid large samples for the training of machine learning algorithms, as well as a platform for independent evaluation of algorithm output (figure 1).

The data analysed consisted of deidentified text from all inpatient and outpatient clinical notes and correspondence. The algorithms used were applied to text

surrounding relevant keywords corresponding to each variable of interest; these keywords were defined by the authors to ensure detecting as wide a range of formulations as possible. For example, to detect free-text mentions of passivity delusions, the keyword "passiv*" was used, with the asterisk allowing for different formulations such as "passive" or "passivity". The documents identified were then manually annotated via the TextHunter platform to train the algorithm to automatically determine if this was relevant; for example, text stating that 'the patient described passive external control of actions' would be marked as positive, text stating 'the patient described a passive lifestyle' would be marked as irrelevant, and text stating 'no passivity' would be marked as a negation statement. Once the algorithm had been optimised, independent evaluation took place on novel text to ascertain precision (proportion of algorithm-identified positive instances found to be correct on evaluation) and recall (proportion of positive entities identified as such by the algorithm) statistics. Testing the performance of the algorithm over the entire CRIS database was performed by checking 100 random annotated documents produced by the algorithm and 100 random unannotated documents using keyword searches. this constituted a gold-standard set from which a second researcher double annotated a proportion of the retrieved documents to determine inter-rater reliability using Cohen's kappa. The process was iterative and sought to achieve P and R values over 80%. High-performance metrics were attained (online supplemental table 1). For primary analyses, positive indications of the three symptoms representing thought interference (thought insertion, thought withdrawal, thought broadcast) were combined under that a single entity with secondary analyses considering them separately; all symptoms were defined as binary variables according to their recording or not in the first 3 months after first presentation.

## Outcomes

Occurrence of negative outcomes was recorded over 2 years following first presentation at SLaM. Three outcomes were evaluated: (1) highest-quartile time spent as an inpatient or receiving support from a home treatment team in the community since first presentation (defined as 22 days or more); (2) receiving more than two antipsychotics and (3) involuntary psychiatric hospitalisation defined as detention for assessment or treatment under sections 2 or 3 of the UK Mental Health Act (MHA).

## Covariates

Included covariates were as follows: gender, ethnic group, age at first SLaM presentation, diagnosis, Index of Multiple Deprivation (IMD) score and presence of recorded paranoia, persecutory delusions or auditory hallucinations. Age was categorised by decades for descriptive analyses but applied as a continuous variable otherwise. Diagnoses were recorded as a binary variable,

distinguishing schizophrenia (ICD-10 F20x) from other schizophreniform disorders (F21–F29).[33] The IMD score is a measure of neighbourhood deprivation, applied at lower super output area level (a standard national address code containing an average 1500 residents), and derived from composite national Census measures. This was divided in quartiles ranging from low (1) to high (4) deprivation. The presence of paranoia, auditory hallucinations and persecutory delusions, three common psychotic symptoms in schizophrenia either previously suggested to strongly interact with passivity[20 34] or in the case of paranoia to constitute a syndrome distinct from that characterised by the presence of hallucinations or other delusions,[24] was also ascertained for adjustment, using previously developed NLP algorithms.[31]

## Statistical analysis

All analyses were carried out using Stata V.15.1 software. Initial descriptive analyses assessed overlap between symptoms, demographic/clinical factors and other psychotic symptoms. After describing the prevalence of outcomes by covariate, logistic regression models were assembled to investigate associations and their independence in relation to each binary outcome. Adjusted R-squared was used to assess model fit. Secondary analyses studied the impact of adjusting for covariates individually to clarify potential confounding effects. Finally, Poisson regression models investigated the relationship between number of negative outcomes and number of symptoms experienced as well as between number of negative outcomes and each individual symptom. Additional linear regression was carried out to study the relationship between symptoms and number of antipsychotics prescribed, assessing the reliability of the latter as a marker of negative outcome.

## Patient and public involvement

The study proposal was reviewed and approved by the patient-led CRIS oversight committee prior to project commencement. No other consultations were made with patients or the public.

## RESULTS

Table 1 describes symptom prevalence levels by demographic and clinical groups. Thought interference and somatic passivity were more common in younger patients and marginally higher in men compared with women but showed no consistent associations with ethnicity or neighbourhood deprivation. Both symptoms occurred more often in the presence of other psychotic symptoms but did not differ substantially between comparison diagnostic groups. Descriptive results indicated that the symptoms of interest overlapped. This was further analysed by describing the co-occurrence of symptoms and later analysing the correlation between this and negative outcomes. Degree of overlap between the subcomponents of thought interference are summarised in online supplemental tables 2 and 3 describes overlap between

**Table 1** Symptom prevalence by demographic characteristics and diagnoses, and overlap with other psychotic symptoms

| | None (%) | Interference without passivity (%) | Broadcast (%) | Insertion (%) | Withdrawal (%) | Passivity without interference (%) | Interference and passivity (%) | Total |
|---|---|---|---|---|---|---|---|---|
| Total participants | 77.0 | 12.8 | 10.3 | 10.7 | 4.9 | 4.9 | 5.2 | 9323 |
| Age at diagnosis | | | | | | | | |
| 16–25 | 65.8 | 19.3 | 16.1 | 17.4 | 7.8 | 6.3 | 8.5 | 2131 |
| 26–35 | 72.4 | 14.8 | 12.7 | 12.5 | 6.0 | 5.9 | 6.9 | 2457 |
| 36–45 | 80.2 | 11.3 | 8.6 | 8.7 | 4.4 | 4.3 | 4.2 | 1799 |
| 46–55 | 82.8 | 10.2 | 6.9 | 8.1 | 3.6 | 3.8 | 3.2 | 1241 |
| 56–65 | 87.6 | 6.9 | 4.1 | 4.8 | 2.0 | 3.8 | 1.7 | 651 |
| 66–75 | 91.0 | 4.8 | 3.0 | 4.0 | 0.5 | 2.6 | 1.6 | 568 |
| 76–85 | 93.5 | 3.4 | 2.6 | 1.6 | 1.0 | 2.9 | 0.3 | 383 |
| 86–95 | 92.5 | 6.5 | 5.4 | 2.2 | 0.0 | 1.1 | 0.0 | 93 |
| Gender | | | | | | | | |
| Female | 80.3 | 11.2 | 8.5 | 10.2 | 3.8 | 4.4 | 4.2 | 3920 |
| Male | 74.7 | 14.1 | 11.5 | 11.1 | 5.7 | 5.3 | 6.0 | 5399 |
| Ethnicity | | | | | | | | |
| Black | 74.9 | 14.1 | 11.2 | 11.7 | 5.5 | 5.4 | 5.7 | 3065 |
| South Asian | 76.1 | 16.3 | 12.0 | 11.6 | 7.3 | 2.7 | 5.0 | 301 |
| Chinese | 72.4 | 9.2 | 10.5 | 7.9 | 2.6 | 10.5 | 7.9 | 76 |
| Other Asian | 72.7 | 13.3 | 13.7 | 12.3 | 6.8 | 3.8 | 10.2 | 293 |
| White | 78.3 | 12.4 | 10.1 | 10.3 | 4.4 | 4.6 | 4.8 | 3631 |
| Other/unknown | 79.0 | 11.4 | 8.3 | 9.6 | 4.3 | 5.0 | 4.6 | 1957 |
| Index of Multiple Deprivation (IMD) | | | | | | | | |
| IMD quartile 1 | 78.1 | 12.7 | 10.5 | 9.9 | 4.7 | 4.4 | 4.8 | 2331 |
| IMD quartile 2 | 77.5 | 12.1 | 9.7 | 10.3 | 4.5 | 5.1 | 5.3 | 2331 |
| IMD quartile 3 | 75.7 | 13.1 | 10.5 | 11.7 | 5.2 | 5.3 | 5.8 | 2338 |
| IMD quartile 4 | 76.8 | 13.4 | 10.3 | 10.8 | 5.4 | 4.8 | 5.0 | 2323 |
| Diagnosis | | | | | | | | |
| Schizophrenia diagnosis (F20) | 78.2 | 12.0 | 9.8 | 9.5 | 4.7 | 4.9 | 4.9 | 3475 |
| Schizotypal/non-mood psychotic disorder (F21–29) | 75.3 | 13.4 | 10.5 | 11.4 | 5.0 | 4.9 | 6.5 | 5848 |
| Other psychotic symptom | | | | | | | | |
| Paranoia present | 73.2 | 15.0 | 12.2 | 12.5 | 5.7 | 5.7 | 6.2 | 7478 |
| Paranoia absent | 92.7 | 4.5 | 2.5 | 3.5 | 1.8 | 1.7 | 1.1 | 1845 |
| Auditory hallucinations present | 65.8 | 18.4 | 16.2 | 16.7 | 7.3 | 6.9 | 9.0 | 4971 |
| Auditory hallucinations absent | 89.9 | 6.5 | 3.5 | 3.8 | 2.2 | 2.6 | 1.0 | 4352 |
| Persecutory delusions present | 66.2 | 18.1 | 15.6 | 16.4 | 7.0 | 7.3 | 8.4 | 4375 |
| Persecutory delusions absent | 86.0 | 8.2 | 5.6 | 5.6 | 3.1 | 2.8 | 2.4 | 4984 |

passivity, thought interference and the other positive symptoms included as covariates.

Using logistic regression, ORs were calculated to further study overlap between symptoms (online supplemental table 4), reiterating increased prevalence of thought interference and somatic passivity when other symptoms are present. Thought insertion and broadcasting were strongly correlated (OR=11.67) as were insertion and withdrawal (OR=12.10), whereas there was a weaker association between broadcast and withdrawal (OR=6.05). Somatic passivity was relatively strongly associated with broadcast (OR=5.13) and insertion (OR=6.66) but was more strongly associated with other psychotic symptoms such as paranoia (OR=4.67) and auditory hallucinations (OR=4.99) than with thought withdrawal (OR=3.86).

Table 2 describes outcome distributions. Younger patients, patients from black ethnic groups and patients from higher deprivation neighbourhoods were at greater risk of negative outcomes. Patients with schizophrenia were less likely to have an MHA section or higher time in crisis care than those with other diagnoses but more likely to be prescribed multiple antipsychotics. Negative outcomes were higher in patients with reported paranoia, auditory hallucinations or persecutory delusions and in those with thought interference or somatic passivity. As mentioned previously, based on past descriptions of symptoms as co-occurring, we described their overlap in online supplemental tables 2 and 3. Similarly, we investigated whether our outcomes of interest occurred independently or together. Online supplemental table 5 describes overlap between outcomes.

Table 3 summarises logistic regression analyses for any negative outcome. Unadjusted models showed significant associations with somatic passivity and all components of thought interference, and these remained significant after adjustments except thought broadcast. Although confidence intervals overlapped, associations with somatic passivity were consistently strongest. Post hoc investigations of those with non-overlapping somatic passivity and thought interference showed a significant association with somatic passivity (OR 1.57, 95% CI 1.27 to 1.95) but not with thought interference (OR 1.11, 95% CI 0.97 to 1.27).

Table 4 displays further analyses of individual outcomes. Associations with somatic passivity remained significant for all after adjustments. Thought interference and insertion specifically were independently associated with higher time spent in crisis care and greater numbers of antipsychotics used, but not with MHA sections. Thought broadcast specifically was not independently associated with any outcome, and thought withdrawal was only associated with higher time spent in crisis care and MHA sections. Aside from thought withdrawal, all other symptoms had the weakest association with MHA sections.

Table 5 summarises analyses investigating adjustments by individual covariates. For both somatic passivity and thought interference, strongest reductions in OR occurred following adjustments for age and the other

three psychotic symptoms; however, OR for somatic passivity remained stronger throughout. Estimated marginal means (EM) were calculated to determine the likelihood of negative outcomes at different ages (online supplemental table 6); this showed a decrease in risk inversely proportional to age: for a patient with somatic passivity, at age 20 EM=59.6% (58.0%–61.1%); at age 90, EM%=21.5% (19.2%–23.8%). For a patient with thought interference, at age 20 EM=59.1% (57.6%–60.7%); at age 90, EM=22.2% (19.8%–24.6%). Finally, exploratory testing investigating the cumulative relationship between symptoms and outcomes, calculating the odds of experiencing multiple negative outcomes in relation to the sum of individual symptoms and to each individual symptom and illustrated using EM, described a 14.9% increase in the risk of multiple negative outcomes when experiencing no versus all four symptoms, from 73.1% to 88.0%. Consistently with identified trends, somatic passivity was the individual symptom with the greatest impact (adjusted OR=1.16 (95% CI 1.09 to 1.23)) and thought broadcasting had no significant effect after adjustment (OR=0.99 (95% CI 0.92 to 1.05)). Regression analysis identified a significant interaction between somatic passivity and number of antipsychotics (p<0.05; adjusted R-squared=0.1031). Thought interference and its subcomponents, also included in the model, had no significant effect (online supplemental table 7).

## DISCUSSION

Having developed NLP algorithms to extract recorded thought interference and somatic passivity in routine electronic mental health records, we investigated their associations with negative outcomes over a 2-year period following first-recorded diagnosis of schizophrenia or a schizophreniform disorder. Both symptoms were associated with a significantly increased risk of experiencing negative outcomes, somatic passivity having the strongest associations after full adjustment. This is broadly consistent with previous reports that experiencing these symptoms is predictive of negative outcomes: Stephen et al[16] found the absence of somatic passivity to correlate with better outcomes in 66% of patients; Koehler[17] found no significant difference in outcomes between patients experiencing any thought interference subcomponent or somatic passivity but concluded that their absence was predictive of a better prognosis. Malinowski et al found that thought broadcasting seemed predictive of disease continuation without remission, but that this effect was no longer significant after adjusting for external factor influence (specifically, untreated psychosis duration and drug abuse)[35]; accordingly, although different covariates were included, our results suggest that thought broadcast does not have a significant effect on outcome. Similar prevalence and overlap between thought insertion, withdrawal and broadcasting suggests that this cannot be due to broadcasting occurring more frequently in isolation. It is interesting to note that aside from broadcasting,

**Table 2** Two-year incidence of negative outcomes by symptom and covariate groups

| Covariate | Any negative outcome (%) | MHA section (%) | >2 antipsychotics prescribed (%) | >22 days spent in crisis care (%) | Total participants |
|---|---|---|---|---|---|
| Interference/ passivity | | | | | |
| None | 42.7 | 31.8 | 21.9 | 10.6 | 7181 |
| Interference without passivity | 61.4 | 46.5 | 31.6 | 32.6 | 1198 |
| Broadcast | 63.3 | 47.4 | 34.2 | 34.9 | 956 |
| Insertion | 67.4 | 49.9 | 36.6 | 38.1 | 997 |
| Withdrawal | 66.9 | 52.1 | 36.2 | 40.1 | 459 |
| Passivity without interference | 68.2 | 49.8 | 36.4 | 35.7 | 456 |
| Interference and passivity | 72.3 | 54.5 | 42.2 | 43.2 | 488 |
| Total participants | 47.9 | 35.8 | 24.9 | 25.0 | 9323 |
| Age at diagnosis | | | | | |
| 16–25 | 61.0 | 48.6 | 32.7 | 36.8 | 2131 |
| 26–35 | 52.8 | 40.7 | 27.0 | 26.1 | 2457 |
| 36–45 | 55.3 | 32.9 | 22.5 | 21.2 | 1799 |
| 46–55 | 54.1 | 31.7 | 23.8 | 20.9 | 1241 |
| 56–65 | 33.5 | 21.8 | 17.8 | 16.9 | 651 |
| 66–75 | 29.9 | 17.8 | 15.7 | 16.2 | 568 |
| 76–85 | 24.3 | 15.4 | 11.7 | 13.3 | 383 |
| 86–95 | 24.7 | 14.0 | 14.0 | 12.9 | 93 |
| Gender | | | | | |
| Female | 46.9 | 35.2 | 25.1 | 24.2 | 3920 |
| Male | 48.6 | 36.2 | 24.8 | 25.6 | 5399 |
| Ethnicity | | | | | |
| Black | 57.5 | 47.6 | 30.0 | 31.3 | 3065 |
| South Asian | 43.5 | 30.2 | 24.9 | 19.3 | 301 |
| Chinese | 50.0 | 44.7 | 15.8 | 21.1 | 76 |
| Other Asian | 48.1 | 36.2 | 27.3 | 23.9 | 293 |
| White | 43.9 | 31.1 | 23.3 | 22.1 | 3631 |
| Other/unknown | 40.7 | 26.4 | 20.0 | 21.7 | 1957 |
| Index of Multiple Deprivation (IMD) | | | | | |
| IMD quartile 1 | 45.6 | 32.9 | 24.5 | 23.1 | 2331 |
| IMD quartile 2 | 46.3 | 33.6 | 25.0 | 24.2 | 2331 |
| IMD quartile 3 | 50.3 | 38.1 | 25.8 | 26.6 | 2338 |
| IMD quartile 4 | 49.2 | 38.4 | 24.3 | 26.1 | 2323 |
| Diagnosis | | | | | |
| Schizophrenia diagnosis (F20) | 44.4 | 30.6 | 29.9 | 22.2 | 3475 |
| schizotypal/non-mood psychotic disorder (F21–29) | 49.9 | 38.8 | 22.0 | 26.7 | 5848 |
| Other symptoms | | | | | |
| Paranoia present | 53.9 | 41.0 | 27.7 | 28.9 | 7478 |
| Paranoia absent | 23.6 | 14.5 | 13.5 | 9.4 | 1845 |
| Auditory hallucinations present | 58.2 | 43.8 | 31.3 | 31.7 | 4971 |
| Auditory hallucinations absent | 36.0 | 26.6 | 17.6 | 17.3 | 4352 |
| Persecutory delusions present | 62.2 | 49.2 | 31.9 | 34.4 | 4375 |

**Table 2** Continued

| Covariate | Any negative outcome (%) | MHA section (%) | >2 antipsychotics prescribed (%) | >22 days spent in crisis care (%) | Total participants |
|---|---|---|---|---|---|
| Persecutory delusions absent | 35.7 | 23.7 | 18.6 | 16.6 | 4984 |

MHA, Mental Health Act.

these symptoms remained significantly associated with negative outcomes after including common non-FRS of schizophrenia in our covariates, namely paranoia, auditory hallucinations and persecutory delusions, suggesting the importance of their link with negative outcomes. Secondary testing identified a unique significant effect of somatic passivity on number of antipsychotics prescribed, possibly reflecting poorer response leading to worsened outcomes; importantly, this outcome may reflect a degree of treatment resistance not investigated here but likely to influence patient prognosis. Our findings suggest that somatic passivity and thought interference subcomponents correlate with an increased risk of experiencing negative outcomes, some more strongly than others. In contrast, it is interesting to note that no particular outcome appeared to be disproportionately correlated with symptoms. Although the OR of an MHA section being in place were less strong than other outcomes, this reflects that their use is reserved for the most unwell patients who refuse or lack capacity to accept voluntary community treatment. Overall, the odds of all three outcomes increased in a similar fashion in the presence of thought interference and somatic passivity, reflecting a generally worsened prognosis.

Nuevo et al[18] described schizophrenia symptoms as a continuum whereby the experience of multiple symptoms predicts a worse prognosis. Our finding that patients with either or both thought interference and somatic passivity were more likely to experience negative outcomes than those for whom these symptoms were absent supports this. Exploratory analyses suggested a significant relationship between number of symptoms experienced and negative outcome occurrence; however, our results also suggest that each symptom has a different impact on outcomes which was not always individually significant, and that the presence of multiple symptoms may even sometimes reduce the frequency of negative outcomes: for example, patients experiencing somatic passivity only may have poorer outcomes than those with both this and thought interference. One explanation may be that reporting thought interference to clinicians requires greater insight from the patient, contributing to better outcomes and facilitating voluntary treatment such as that offered by crisis care teams with which more patients experiencing thought insertion and withdrawal engaged. The recording of exposure variables relied on symptoms being elicited and noted by clinicians in routine practice and therefore cannot be assumed to be equivalent to those ascertained in a more research-focused interview, just as covariates such as diagnosis also were derived from routine clinical records and should not be assumed to equate research diagnostic criteria; however, we feel that inaccuracies in exposure ascertainment were more likely to obscure findings of interest through non-differential measurement error rather than give rise to spurious associations. Unfortunately, although mandatory hospitalisation under the

**Table 3** Unadjusted and adjusted OR for the occurrence of any negative outcome in the presence of thought interference and somatic passivity

| Outcome: any negative outcome | Unadjusted OR (95% CI) | Adjusted* OR (95% CI) | Adjusted model pseudo R-squared |
|---|---|---|---|
| Somatic passivity | 2.86 (2.47 to 3.31) | 1.61 (1.37 to 1.88) | 0.12 |
| Interference | 2.30 (2.07 to 2.57) | 1.25 (1.11 to 1.41) | 0.12 |
| Broadcast | 2.01 (1.75 to 2.31) | 1.05 (0.91 to 1.22) | 0.12 |
| Insertion | 2.47 (2.15 to 2.84) | 1.24 (1.15 to 1.55) | 0.12 |
| Withdrawal | 2.29 (1.88 to 2.79) | 1.36 (1.10 to 1.69) | 0.12 |
| Interference or passivity | 2.54 (2.29 to 2.80) | 1.30 (1.19 to 1.42) | 0.12 |
| Both interference and passivity | 1.73 (1.57 to 1.92) | 1.24 (1.11 to 1.38) | 0.12 |
| Post hoc: passivity without interference | 2.44 (1.99 to 2.98) | 1.57 (1.27 to 1.95) | 0.12 |
| Post hoc: interference without passivity | 1.88 (1.66 to 2.13) | 1.11 (0.97 to 1.27) | 0.12 |

*Adjusted for age at diagnosis, gender, ethnicity, diagnosis, IMD quartile and presence of paranoia, auditory hallucinations, persecutory delusion.
IMD, Index of Multiple Deprivation.

**Table 4** Unadjusted and adjusted ORs for associations of each negative outcome with the presence of somatic passivity, thought interference, broadcasting, insertion and withdrawal

| Exposure: somatic passivity | Unadjusted OR (95% CI) | Adjusted* OR (95% CI) | Adjusted model pseudo R-squared |
|---|---|---|---|
| Any negative outcome | 2.86 (2.47 to 3.31) | 1.61 (1.37 to 1.88) | 0.12 |
| >22 days spent in crisis care | 2.15 (1.87 to 2.48) | 1.33 (1.15 to 1.54) | 0.08 |
| MHA section | 2.13 (1.86 to 2.44) | 1.23 (1.06 to 1.42) | 0.13 |
| >2 antipsychotics prescribed | 2.14 (1.86 to 2.46) | 1.43 (1.23 to 1.66) | 0.07 |
| Exposure: thought interference | | | |
| Any negative outcome | 2.30 (2.07 to 2.57) | 1.25 (1.11 to 1.41) | 0.12 |
| >22 days spent in crisis care | 1.89 (1.69 to 2.12) | 1.13 (1.00 to 1.28) | 0.08 |
| MHA section | 1.95 (1.75 to 2.17) | 1.08 (0.96 to 1.21) | 0.13 |
| >2 antipsychotics prescribed | 1.80 (1.60 to 2.01) | 1.16 (1.02 to 1.31) | 0.07 |
| Exposure: thought broadcasting | | | |
| Any negative outcome | 2.01 (1.75 to 2.31) | 1.05 (0.91 to 1.22) | 0.12 |
| >22 days spent in crisis care | 1.71 (1.49 to 1.97) | 1.01 (0.87 to 1.18) | 0.08 |
| MHA section | 1.71 (1.50 to 1.96) | 0.92 (0.80 to 1.07) | 0.13 |
| >2 antipsychotics prescribed | 1.66 (1.44 to 1.91) | 1.05 (0.90 to 1.22) | 0.07 |
| Exposure: thought insertion | | | |
| Any negative outcome | 2.47 (2.15 to 2.84) | 1.34 (1.15 to 1.56) | 0.12 |
| >22 days spent in crisis care | 2.01 (1.75 to 2.31) | 1.21 (1.05 to 1.40) | 0.08 |
| MHA section | 1.93 (1.69 to 2.20) | 1.06 (0.92 to 1.23) | 0.13 |
| >2 antipsychotics prescribed | 1.88 (1.64 to 2.16) | 1.22 (1.05 to 1.41) | 0.07 |
| Exposure: thought withdrawal | | | |
| Any negative outcome | 2.29 (1.88 to 2.79) | 1.36 (1.10 to 1.69) | 0.12 |
| >22 days spent in crisis care | 2.09 (1.73 to 2.54) | 1.37 (1.12 to 1.68) | 0.08 |
| MHA section | 2.02 (1.68 to 2.44) | 1.23 (1.01 to 1.51) | 0.13 |
| >2 antipsychotics prescribed | 1.76 (1.45 to 2.14) | 1.21 (0.98 to 1.48) | 0.07 |

*Adjusted for age at diagnosis, gender, ethnicity, diagnosis, IMD quartile and presence of paranoia, auditory hallucinations, persecutory delusion.
IMD, Index of Multiple Deprivation; MHA, Mental Health act.

MHA reflects a need for crisis care, it is not possible with these algorithms to determine what interventions patients received and responded to in that context specifically. However, guideline consistency since 2007 suggests that patients represented in this analysis should have received similar standards of care across the time period sampled.

**Table 5** Logistic regression analysis of associations with any negative outcome adjusting separately for individual covariates

| Outcome: any negative outcome | Somatic passivity | Thought interference |
|---|---|---|
| Unadjusted | 2.86 (2.47–3.31) | 2.30 (2.07–2.57) |
| Adjusted for age at diagnosis | 2.47 (2.13–2.86) | 1.94 (1.73–2.17) |
| Adjusted for gender | 2.85 (2.46–3.30) | 2.30 (2.06–2.57) |
| Adjusted for ethnicity | 2.85 (2.46–3.30) | 2.28 (2.04–2.55) |
| Adjusted diagnosis | 2.86 (2.47–3.31) | 2.30 (2.06–2.56) |
| Adjusted for IMD quartile | 2.86 (2.47–3.31) | 2.30 (2.06–2.57) |
| Adjusted paranoia | 2.43 (2.09–2.82) | 1.94 (1.74–2.17) |
| Adjusted for auditory hallucinations | 2.23 (1.92–2.59) | 1.80 (1.61–2.02) |
| Adjusted for persecutory delusions | 2.23 (1.91–2.59) | 1.82 (1.63–2.04) |

IMD, Index of Multiple Deprivation.

Overall, although there does seem to be a trend of worsening outcomes as symptoms accumulate, it would be overly simplistic to claim that this effect is purely additive.

Interestingly, of the covariates investigated, only age had a substantial confounding effect, negatively correlated with symptoms and outcomes of interest. In the past, thought insertion and withdrawal have been reported as more common in early-onset cases and somatic passivity as more frequent in late-onset cases.[11] Because of challenges in establishing whether patients were previously diagnosed in another service before presenting at SLaM, it is not possible to equate first presentation here with first-episode psychosis, particularly in older individuals who may have longer standing diagnoses and recently in-migrated to the catchment. This first presentation sample is, therefore, better viewed as one which is enriched with first-episode cases rather than directly equivalent. Furthermore, as our database consists of real-life data patients do not always have consistent follow-up or repeat assessments, including due to external factors not related to the patient's care. Our model is, therefore, useful in correlating patient outcomes with symptoms at first presentation but it is not possible to infer stability of the clinical construct over time as the patient is only assessed once. Although age at diagnosis may not accurately reflect age at disease onset and doesn't account for possible changes in symptom profile as patients age, our results suggest that both symptoms are more frequent in younger patients, leading to a worsened prognosis. In this context, it would be interesting, in future research, to investigate symptom stability over time and how prognosis might also change with this.

Other external factors such as level of education or spiritual beliefs may play a role in the personal interpretation of abnormal experiences and have consequently affected outcomes. We developed a model that suggested the existence of significant interactions between thought interference, somatic passivity and the occurrence of negative outcomes. As shown in online supplemental tables 2 and 3, these symptoms overlapped in many patients as well as with other psychotic symptoms, and so did the negative outcomes under scrutiny. The overlap observed before further analyses suggested that there may exist interactions linking these factors and directed our analyses towards the development of a model to understand these better. The algorithms we developed to create this model achieved high-performance metrics and our model-fit measures demonstrate model reliability but highlight that the variance remains in great part unexplained, likely due to the role of other factors not included here. This could be usefully explored in further studies to refine models assessing patient prognosis. Furthermore, our patient sample was drawn from an inner-urban South-London population and results may be different in other patient groups, although the large sample size should mitigate this effect.

Paranoia, persecutory delusions and auditory hallucinations were included as covariates due to past evidence suggesting a correlation with thought interference and somatic passivity. The primary purpose of this study was to characterise symptoms of thought interference and passivity, and other symptoms, such as negative and disorganised symptoms, were not included. Broader symptom profiles should be investigated in future research. Interactions between outcomes should also be studied; for example, it is not unreasonable to propose that the prescription of fewer antipsychotics may correlate with reduced hospitalisation time. To this effect, a harmonisation of definitions would be useful to enable the development of algorithms studying symptoms described more heterogeneously. Outcomes analysed were restricted to those most readily available from the source data and most likely to be clinically informative; however, they cannot be viewed as exhaustive and there may be elements of prognosis that were not captured in this study, including interactions between outcomes and previous adverse events that may influence symptom occurrence and recording.

Although to be treated with caution, as not tested for significance, our descriptive results provide potential directions for future research and support the notion that psychotic symptoms share more complex interactions than previously described. Somatic passivity has been suggested to cluster with different symptoms such as persecutory delusions[4] or auditory-verbal hallucinations,[5] and thought interference has sometimes been described as a component of somatic passivity.[18 36] We found that subcomponents of thought interference traditionally assumed to closely correlate may actually more commonly occur alongside other symptoms; for example, the odds of experiencing thought insertion alongside somatic passivity were greater than those of withdrawal and broadcasting co-occurring. In a study attempting to link delusions to putative underlying neurobiological mechanisms, Kimhy *et al* called for new symptom definitions that would focus on underlying mechanisms rather than phenomenological descriptions.[23] Each delusion was described then grouped with others based on correlation, enabling the grouping of symptoms that share significant interactions while maintaining their individuality. This approach enables a less biased study of psychotic symptoms relying less on traditional groupings and may allow more accurate descriptions of symptom interactions. Additionally, thought interference and somatic passivity have previously been found to be closely related and predictive of schizophrenia.[13 36–38] Our results support reports that these symptoms share a positive correlation[20] but question their specificity to schizophrenia and usefulness in differentiating this from other psychotic disorders, particularly as both were absent in 77% of patients. Somatic passivity has been reported to be the most common symptom experienced in schizophrenia[6] and first-episode psychosis[24]; however, in our study thought interference was almost twice as commonly recorded. Overall, it would be interesting to repeat these analyses in the future with a broader range of diagnoses including affective psychoses in which they also occur to determine whether these

symptoms could be more specific to other conditions or used to assess patient prognosis.

## CONCLUSION

This study developed NLP applications to detect the recorded presence of somatic passivity, thought interference and its subcomponents in patients within 3 months of their first presentation and analysed negative outcome occurrence over the two following years to investigate the relationship between these specific FRS and patient prognosis. We observed a significant effect of somatic passivity, thought interference and its subcomponents thought insertion and withdrawal on the likelihood of experiencing any negative outcome. We also offer evidence that psychotic symptoms traditionally associated with schizophrenia may not be specific to this disorder and share complex interactions, putting in question current definitions and warranting further investigation to determine whether it is correct to consider thought insertion, withdrawal and broadcasting as parts of the same symptom. Overall, the presence of somatic passivity and thought interference was found to significantly increase the risk of experiencing negative outcomes in psychosis; somatic passivity had the greatest effect. Future research should focus on acquiring a better understanding of these effects, including by investigating other psychotic symptoms to determine the more detailed pathways through which these interact with prognosis in schizophrenia and other types of psychosis. Future studies should investigate these correlations and reliable markers of outcomes to develop standardised tools enabling a better evaluation of patient prognosis. In addition, it would be interesting to study which antipsychotics specifically were prescribed and whether patients may have had a better response to specific medication, reflective of a general lack of response to individual antipsychotics in schizophrenia rather than the presence of refractory disease.

**Contributors** TTM, AK, JS, RP and RS planned the study together. JS developed the applications, which were doubly annotated by TTM and AK. JS and AK retrieved the data, and TTM analysed it with advice from AK, JS, RP and RS. TTM wrote the largest part of the paper, produced the tables and figures and submitted it, working under supervision from AK and RS. AK, RP and RS commented on and directly contributed to the manuscript. RS acts as guarantor for this study.

**Funding** RS, RP and AK are part funded by the National Institute for Health Research (NIHR) Biomedical Research Centre at the South London and Maudsley NHS Foundation Trust and King's College London. RS is additionally part funded by: (1) an NIHR Senior Investigator Award; (2) the National Institute for Health Research (NIHR) Applied Research Collaboration South London (NIHR ARC South London) at King's College Hospital NHS Foundation Trust; (3) the DATAMIND HDR UK Mental Health Data Hub (MRC grant MR/W014386). RP has received funding from an NIHR Advanced Fellowship (NIHR301690), a Medical Research Council (MRC) Health Data Research UK Fellowship (MR/S003118/1) and a Starter Grant for Clinical Lecturers (SGL015/1020) supported by the Academy of Medical Sciences, The Wellcome Trust, MRC, British Heart Foundation, Arthritis Research UK, the Royal College of Physicians and Diabetes UK.

**Disclaimer** The views expressed are those of the authors and not necessarily those of the NIHR or the Department of Health and Social Care.

**Competing interests** RS declares research support received in the last 36 months from Janssen, GSK and Takeda. RP has received grant funds from Janssen and consultancy fees from Induction Healthcare and Holmusk.

**Patient and public involvement** Patients and/or the public were involved in the design, or conduct, or reporting, or dissemination plans of this research. Refer to the Methods section for further details.

**Patient consent for publication** Not applicable.

**Ethics approval** The source deidentified data are made available for secondary analyses as approved by Oxford Research Ethics Committee C, reference 18/SC/0372.

**Provenance and peer review** Not commissioned; externally peer reviewed.

**Data availability statement** No data are available.

**ORCID iDs**
Thibault Thierry Magrangeas http://orcid.org/0000-0002-7019-6091
Anna Kolliakou http://orcid.org/0000-0003-1234-4129
Rashmi Patel http://orcid.org/0000-0002-9259-8788

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
