## [Reviewer comments · BMJ Open]

ARTICLE DETAILS

TITLE (PROVISIONAL)	Investigating the relationship between thought interference, somatic passivity and outcomes in patients with psychosis: a natural-language processing approach using a clinical records search platform in south London
AUTHORS	Magrangeas, Thibault; Kolliakou, Anna; Sanyal, Jyoti; Patel, Rashmi; Stewart, Robert

VERSION 1 – REVIEW

REVIEWER	Tang, Sunny X Northwell Health
REVIEW RETURNED	17-Oct-2021

GENERAL COMMENTS	Thank for the opportunity to review this manuscript by Magrangeas et al, entitled, "Investigating the influence of Thought Interference and Somatic Passivity on outcomes in patients with Psychosis: a Natural-Language Processing (NLP) approach using the Clinical Record Interactive Search (CRIS) platform. The authors used NLP methods to identify patients from a large electronic health database who experienced thought interference and somatic passivity symptoms within 3 months of their initial encounter. Presence of these and related symptoms were related to negative outcomes within 2 years. They found that both types of first rank symptoms conferred higher odds of poor outcomes, most of which continued to be true when accounting for potentially confounding variables. Overall, this manuscript is well-written, and addresses an interesting question with significant potential clinical implications. However, several points should be addressed: 1. The basic premise of investigating the effect of first rank symptoms on clinical outcomes in psychosis is interesting. Why did the authors choose to look at these symptoms, while excluding other first rank symptoms, such as auditory hallucinations of multiple conversing voices, commenting voices, etc.?2. Given the emphasis on NLP in the title and elsewhere, there were few details on the actual NLP process. Please explain the NLP process in greater detail, either in the main article or in supplement. E.g. What keywords were included? How were negation and relevance identified? A flowchart is one way to clarify the NLP pipeline.3. The authors should consider mentioning first rank symptoms in the objective section of the abstract as this gives critical context for why this study is being conducted.4. Did the authors perform any kind of stability analysis to see how consistently these symptoms were identified for individual
--

	patients? For example, were these symptoms mentioned in single records, or in multiple records across episodes? If a standard for consistency is applied, does it strengthen some of the findings – and do more of the findings remain consistent across covarying for confounding variables? 5. Though the first rank symptoms were originally described in the context of schizophrenia spectrum disorders, in distinction from affective psychoses, we should not assume that there is necessarily a qualitative difference between SSD and affective psychoses. Why were affective psychoses not included? 6. A wide range of ages was included, but there was no justification for why <16 and >95 were excluded. 7. Please explain if first presentation at SLaM is at all related to first-episode psychosis. I am not familiar with the health system described here. Based on the ages, I'm guessing that first presentation is NOT the same as first-episode, and therefore older participants are likely to be more chronic patients. 8. Please explain why the 3month cutoff from first presentation. 9. Including other non-first rank symptoms provided additional context for interpreting the findings. Why were negative and disorganized symptoms not included also? 10. Did the authors try covarying for non-first rank symptoms (hallucinations, paranoia), and seeing if the first rank symptoms still predicted the adverse outcomes? 11. Please add a note for the IMD abbreviation in Table 1
--	--

REVIEWER	Levis, Maxwell White River Junction VA Medical Center
REVIEW RETURNED	07-Jan-2022

GENERAL COMMENTS	Thanks for the opportunity to review this compelling manuscript. I found it overall to be very well written. My primary concerns deal with lack of clarity about data type, background on subcomponents, and greater detail about NLP model development. I support publication upon addressing some basic concerns. My specific comments are as follows: Abstract: Could you add bracket meaning for somatic passivity in the same way that you do for interference symptoms? Could you briefly clarify what type of mental healthcare data? Introduction: Could you provide a reference for Schneider's definition of schizophrenia? Not obvious who “the same authors” are. Line 54, pg 4. The articulation about the importance of thought interference is very clear. It would be useful, however, to contextualize the later described subcomponents within this explanation, or at least provide referencing alongside line 56, pg 5. It could be useful to address why auditory hallucinations are not being evaluated as a central part of FRS given the provided supporting literature line 47, pg 5. It would again be useful to clarify what type of mental health data that will be utilized. I could imagine that methods for evaluating clinical notes, outcome measure data, and pathology reports, for example, would all be different. Methods: I am a little confused about difference between first “face-to-face contact” and “first diagnosis”. Again, it would be very helpful to know what type of data is being analyzed.
---

	Sadly, I am not familiar with CRIS, but it sounds like an amazing resource. It would be helpful to know more about algorithm development. Although the authors do address annotation methods and interrater reliability, it would be useful to understand how models were developed and keywords associated with the discrete constructs were identified. While supplementary table 1 is helpful in this regard, this too would be aided by more information and clarifying methods and statistical analysis. For example, I don't sufficiently understand what "manual testing" or "keyword search P and R". Results: Overall, the results are very clear and the usage of tables is effective. A few small notes: "Other diagnosis" (table 1) is a little confusing. I think I understand what the authors are attempting to relay, but this should be clarified. I am not totally clear on how "thought interference only" was measured vis-à-vis the subcomponents. Is it possible to provide basic information on number of sessions, type of care? The Venn diagrams are useful, but supporting literature and explanation in text about model conceptualization would aid argument. Discussion: The presentation is convincing and well argued. Again, some small points: I am curious about the impact of time. Given the study timeframe ranges from 2007-2020, what changes may have impacted assessment and treatment standards? Perhaps some limitation should be added regarding this point? "Commoner" sounds a little odd in American English.
--	--

VERSION 1 – AUTHOR RESPONSE

Reviewer 1:

1. The basic premise of investigating the effect of first rank symptoms on clinical outcomes in psychosis is interesting. Why did the authors choose to look at these symptoms, while excluding other first rank symptoms, such as auditory hallucinations of multiple conversing voices, commenting voices, etc.?

We have answered this in our introduction by emphasising the evidence linking these symptoms together (page 6):

" Recent studies have consistently found thought interference (the specific delusion that thoughts are being inserted in, withdrawn or broadcast from one's head) and somatic passivity (the broader delusion that one's actions and sensations are controlled by an external force) to be core symptoms of psychotic disorders in general and schizophrenia specifically¹⁵⁻¹⁷. (...) Somatic passivity and thought interference were also reported to be positively correlated and predict the occurrence of schizophrenia or schizoaffective disorder²⁰⁻²¹. While some authors have identified differing effects on outcomes, with passivity predicting a worsened prognosis in contrast to interference⁹, others have highlighted a possible link, as both phenomenologically reflect a belief in external control, suggesting

they may have a synergistic effect on outcomes²¹⁻²³ or even constitute components of the same delusion^{5&24}."

We have also discussed the difficulty of developing algorithms for symptoms whose descriptions are more heterogeneous and therefore harder to detect via an automated process (page 7):

". Our focus on thought interference and somatic passivity as particular symptoms aimed to clarify the interactions described previously which suggested a particularly strong relationship between these symptoms; in addition, they were chosen as potentially tractable to extraction at scale from routine healthcare data via NLP, being described in relatively consistent language in clinical records and thus a pragmatic choice for further investigation in contrast to other symptoms such as running commentary hallucinations which are typically described with greater linguistic heterogeneity based on the experience rather than the phenomenology and therefore harder to study with this approach."

2. Given the emphasis on NLP in the title and elsewhere, there were few details on the actual NLP process. Please explain the NLP process in greater detail, either in the main article or in supplement. E.g. What keywords were included? How were negation and relevance identified? A flowchart is one way to clarify the NLP pipeline.

Following this recommendation, we have included a new figure to clarify the NLP pipeline and provided further explanations in the figure legend (page 9):

Figure 1: Flowchart illustrating the development of algorithms via the TextHunter platform. TextHunter allows a training dataset of text strings containing wording of interest to be presented in an easy visual form for researcher annotation. The development process is identical for all algorithms relating to symptoms recorded in clinical text. In the final step, testing the performance of the algorithm over the entire CRIS database was performed by checking 100 random annotated documents produced by the algorithm and 100 random unannotated documents using

keyword searches. A second researcher double-annotated a proportion of the retrieved documents to determine inter-rater reliability (IRR) using Cohen's kappa. The process was iterative and sought to achieve P and R values over 80%.

3. The authors should consider mentioning first rank symptoms in the objective section of the abstract as this gives critical context for why this study is being conducted.

This has now been included in the objectives as suggested (page 2):

" Objectives: We aimed to apply natural language-processing (NLP) algorithms in routine healthcare records to identify reported somatic passivity (external control of sensations, actions and impulses) and thought interference symptoms (thought broadcasting, insertion, withdrawal), First-Rank symptoms traditionally central to diagnosing schizophrenia, and determine associations with prognosis by analysing routine outcomes."

4. Did the authors perform any kind of stability analysis to see how consistently these symptoms were identified for individual patients? For example, were these symptoms mentioned in single records, or in multiple records across episodes? If a standard for consistency is applied, does it strengthen some of the findings – and do more of the findings remain consistent across covarying for confounding variables?

We have discussed this further in our discussion and added a related limitation to explain why it was not possible to carry out stability testing over time (pages 24 and 4 respectively):

"Furthermore, as our database consists of real-life data patients do not always have consistent follow-up or repeat assessments, including due to external factors not related to the patient's care. Our model is therefore useful in correlating patient outcomes with symptoms at first presentation but it is not possible to infer stability of the clinical construct over time as the patient is only assessed once."

5. Though the first rank symptoms were originally described in the context of schizophrenia spectrum disorders, in distinction from affective psychoses, we should not assume that there is necessarily a qualitative difference between SSD and affective psychoses. Why were affective psychoses not included?

We chose to focus specifically on patients suffering from SSD because of the original descriptions here, and we agree that it would be interesting to construct a further study to include affective psychoses although we feel that this is beyond the scope of this paper. Text has been added to the discussion (page 27) on this limitation and need for further research:

" Overall, it would be interesting to repeat these analyses in the future with a broader range of diagnoses including affective psychoses in which they also occur to determine whether these symptoms could be more specific to other conditions or used to assess patient prognosis."

6. A wide range of ages was included, but there was no justification for why <16 and >95 were excluded.

A justification for this has now been added in our methods section (page 8):

" This age-range was chosen to cover patients transitioning to adult mental health care, commonly occurring from age 16; patients with no recorded date of birth appear on the system as aged 100+ and represented all the individuals above age 95; they were therefore excluded to avoid skew when correcting for age."

7. Please explain if first presentation at SLaM is at all related to first-episode psychosis. I am not familiar with the health system described here. Based on the ages, I'm guessing that first presentation is NOT the same as first-episode, and therefore older participants are likely to be more chronic patients.

An explanation for this has also been added to our methods section and our discussion to describe associated limitations (pages 9 and 24 respectively):

" Thought interference and somatic passivity occurrence in the three months following first presentation at SLaM (defined as first face-to-face contact) was sought" and " Because of challenges in establishing whether patients were previously diagnosed in another service before presenting at SLaM, it is not possible to equate first presentation here with first-episode psychosis, particularly in older individuals who may have longer standing diagnoses and recently in-migrated to the catchment. This first presentation sample is therefore better viewed as one which is enriched with first-episode cases rather than directly equivalent."

8. Please explain why the 3month cutoff from first presentation.

We chose this cut-off to ensure that the sample included patients with these symptoms at first presentation rather than developing them later; the 3-month period was felt to provide sufficient time for clinicians to assess all patients and record their symptoms accurately. This consideration has been added to the methods section (page 9): This cut-off was chosen to allow time for clinicians to assess all patients and record their symptoms accurately and focus on patients presenting with these symptoms as first presentation rather than later development.

9. Including other non-first rank symptoms provided additional context for interpreting the findings. Why were negative and disorganized symptoms not included also?

Similar to our response to the first point raised, these symptoms are often described in clinical notes with greater heterogeneity and therefore harder to accurately analyse with automated algorithms. This has been added to the discussion (page 26) alongside the need for more harmonised definitions to enable this analysis:

"To this effect, a harmonisation of definitions would be useful to enable the development of algorithms studying symptoms described more heterogeneously. Outcomes analysed were restricted to those most readily available from the source data and most likely to be clinically informative; however, they cannot be viewed as exhaustive and there may be elements of prognosis that were not captured in this study, including interactions between outcomes and previous adverse events that may influence symptom occurrence and recording."

10. Did the authors try covarying for non-first rank symptoms (hallucinations, paranoia), and seeing if the first rank symptoms still predicted the adverse outcomes?

Non-first rank symptoms were indeed included as covariates, specifically presence of paranoia, auditory hallucinations, persecutory delusion (table 5, page 20). Alongside age at diagnosis, these were the only covariates to have a strong influence on the interaction between somatic passivity and thought interference.

11. Please add a note for the IMD abbreviation in Table 1

This has been added.

Reviewer 2:

Abstract:

Could you add bracket meaning for somatic passivity in the same way that you do for interference symptoms?

Could you briefly clarify what type of mental healthcare data?

Both these suggestions have been followed (page 2):

"Objectives: We aimed to apply natural language-processing (NLP) algorithms in routine healthcare records to identify reported somatic passivity (external control of sensations, actions and impulses) and thought interference symptoms (thought broadcasting, insertion, withdrawal)." and "Design: Four algorithms were developed on de-identified mental healthcare data and applied to ascertain recorded symptoms over the three months following first presentation to a mental healthcare provider in a cohort of patients with a primary schizophreniform disorder (ICD-10 F20-F29) diagnosis."

Introduction:

Could you provide a reference for Schneider's definition of schizophrenia?

This has been added (page 5).

Not obvious who "the same authors" are. Line 54, pg 4.

This has been clarified by placing the reference to a more appropriate place in the sentence (page 6):

"In agreement with these findings, recent DSM-5 changes have been welcomed as a valuable push towards a better understanding of individual symptoms¹⁴; however, the same authors explain that limitations remain as this new focus does not satisfyingly address the relationship between symptoms, conflicts with most existing guidelines for early symptom detection in schizophrenia that heavily rely on FRS, and diverts attention away from core psychopathological processes at the disorder's root."

The articulation about the importance of thought interference is very clear. It would be useful, however, to contextualize the later described subcomponents within this explanation, or at least provide referencing alongside line 56, pg 5.

This has been clarified at the start of the introduction, page 5:

" The latter was subdivided into three types of thought interference depending on whether the patient perceived their thoughts to be removed by an external entity, inserted externally or broadcast to the world."

It could be useful to address why auditory hallucinations are not being evaluated as a central part of FRS given the provided supporting literature line 47, pg 5.

As raised by the first reviewer in point 1, we have answered this in our introduction by emphasising the evidence linking thought interference and somatic passivity symptoms specifically together (page 6), with some authors suggesting them to be components of the same symptom. We also discussed the difficulty of developing algorithms for symptoms whose descriptions are more heterogeneous and therefore harder to detect via an automated process (page 7):

"Our focus on thought interference and somatic passivity as particular symptoms aimed to clarify the interactions described previously which suggested a particularly strong relationship between these symptoms; in addition, they were chosen as potentially tractable to extraction at scale from routine healthcare data via NLP, being described in relatively consistent language in clinical records and thus a pragmatic choice for further investigation in contrast to other symptoms such as running commentary hallucinations which are typically described with greater linguistic heterogeneity based on the experience rather than the phenomenology and therefore harder to study with this approach."

It would again be useful to clarify what type of mental health data that will be utilized. I could imagine that methods for evaluating clinical notes, outcome measure data, and pathology reports, for example, would all be different.

This has been clarified in the abstract and methods (pages 2 and 9). This study relied purely on clinical notes from the CRIS database:

" Design: Four algorithms were developed on de-identified mental healthcare data and applied to ascertain recorded symptoms over the three months following first presentation to a mental healthcare provider in a cohort of patients with a primary schizophreniform disorder (ICD-10 F20-F29) diagnosis" and " The trust provides care to 37,500 active patients, with 12,000 to 14,000 clinical events per week covering a diverse range of interventions in the community and in the inpatient setting^{25&26}. SLAM's Clinical Record Interactive Search (CRIS) platform was developed in 2008 to enable research on a de-identified copy of the Trust's electronic health records (used across all services since 2006) within a robust governance and data security framework²⁴."

Methods:

I am a little confused about difference between first "face-to-face contact" and "first diagnosis". Again, it would be very helpful to know what type of data is being analyzed.

The question regarding first face-to-face contact was also raised by the first reviewer and the answer is the same as point 7. As stated previously, the nature of our data have been, we hope, further clarified in the abstract and methods section. First face-to-face contact will be the first time someone is seen by the clinical service; however, the diagnostic formulation is often not complete at the outset, so diagnosis date is less useful as a measure of onset.

Sadly, I am not familiar with CRIS, but it sounds like an amazing resource. It would be helpful to know more about algorithm development. Although the authors do address annotation methods and

interrater reliability, it would be useful to understand how models were developed and keywords associated with the discrete constructs were identified. While supplementary table 1 is helpful in this regard, this too would be aided by more information and clarifying methods and statistical analysis. For example, I don't sufficiently understand what "manual testing" or "keyword search P and R".

As this was also requested by the previous reviewer in their second point, we have created a flowchart to clarify the algorithm development process and expanded on this in our figure legend.

Results:

"Other diagnosis" (table 1) is a little confusing. I think I understand what the authors are attempting to relay, but this should be clarified.

This has been replaced with "schizotypal/non-mood psychotic disorder (F21-29)" to ensure clarity.

I am not totally clear on how "thought interference only" was measured vis-à-vis the subcomponents.

This wording in the tables has been changed from "thought interference only" to "interference without passivity". This refers to interference occurring by itself with no recorded passivity and was defined to study the differential relationship of these symptoms with patient prognosis.

Is it possible to provide basic information on number of sessions, type of care?

Unfortunately, due to the nature of our database this is difficult to establish, and our analysis was focused on patient presentation and outcomes but could not ascertain different types of care. The nature of care provided at SLaM was expanded on in the methods section (page 9), including a reference describing the services available at SLaM:

" The trust provides care to 37,500 active patients, with 12,000 to 14,000 clinical events per week covering a diverse range of interventions in the community and in the inpatient setting^{25&26}."

This was also added to our discussion and limitations (pages 26 and 4):

" Because of challenges in establishing whether patients were previously diagnosed in another service before presenting at SLaM, it is not possible to equate first presentation here with first-episode psychosis, particularly in older individuals who may have longer standing diagnoses and recently in-migrated to the catchment. This first presentation sample is therefore better viewed as one which is enriched with first-episode cases rather than directly equivalent. Furthermore, as our database consists of real-life data patients do not always have consistent follow-up or repeat assessments, including due to external factors not related to the patient's care. Our model is therefore useful in correlating patient outcomes with symptoms at first presentation but it is not possible to infer stability of the clinical construct over time as the patient is only assessed once" and " Our model is useful in correlating patient outcomes with symptoms at first presentation but it is not possible to infer stability of the clinical construct over time as the patient is only assessed once."

The Venn diagrams are useful, but supporting literature and explanation in text about model conceptualization would aid argument.

Our model we hope is now better explained in our discussion, pages 25-27. We developed a model that suggested the existence of significant interactions between thought interference, somatic passivity, and the occurrence of negative outcomes. As shown in supplementary figures 1 and 2, these symptoms overlapped in many patients as well as with other psychotic symptoms, and so did the negative outcomes under scrutiny. The algorithms we developed to create this model achieved high performance metrics and our model-fit measures demonstrate model reliability but highlight that the variance remains to a large part unexplained, likely due to the role of other factors not included here. This might be usefully explored in further studies to refine models assessing patient prognosis and we have added text to highlight this:

" We developed a model that suggested the existence of significant interactions between thought interference, somatic passivity and the occurrence of negative outcomes. As shown in supplementary figures 1 and 2, these symptoms overlapped in many patients as well as with other psychotic symptoms, and so did the negative outcomes under scrutiny. The algorithms we developed to create this model achieved high performance metrics and our model-fit measures demonstrate model reliability but highlight that the variance remains in great part unexplained, likely due to the role of other factors not included here. This could be usefully explored in further studies to refine models assessing patient prognosis. Furthermore, our patient sample was drawn from an inner-urban South-London population and results may be different in other patient groups, although the large sample size should mitigate this effect.

Paranoia, persecutory delusions and auditory hallucinations were included as covariates due to past evidence suggesting a correlation with thought interference and somatic passivity. The primary purpose of this study was to characterise symptoms of thought interference and passivity, and other symptoms, such as negative and disorganised symptoms, were not included. Broader symptom profiles should be investigated in future research. Interactions between outcomes should also be studied; for example, it is not unreasonable to propose that the prescription of fewer antipsychotics may correlate with reduced hospitalisation time. To this effect, a harmonisation of definitions would be useful to enable the development of algorithms studying symptoms described more heterogeneously. Outcomes analysed were restricted to those most readily available from the source data and most likely to be clinically informative; however, they cannot be viewed as exhaustive and there may be elements of prognosis that were not captured in this study, including interactions between outcomes and previous adverse events that may influence symptom occurrence and recording."

Discussion:

I am curious about the impact of time. Given the study timeframe ranges from 2007-2020, what changes may have impacted assessment and treatment standards? Perhaps some limitation should be added regarding this point?

This has been further mentioned in our discussion (pages 24-25). Guideline consistency since 2007 ensures that patients received similar standards of care and we feel that the same treatment standards are likely to have been maintained, although acknowledge that we cannot directly measure this.

" Unfortunately, although mandatory hospitalisation under the MHA reflects a need for crisis care, it is not possible with these algorithms to determine what interventions patients received and responded to in that context specifically. However, guideline consistency since 2007 suggests that patients represented in this analysis should have received similar standards of care across the time period sampled" and " Although age at diagnosis may not accurately reflect age at disease onset and doesn't account for possible changes in symptom profile as patients age, our results suggest that both

symptoms are more frequent in younger patients, leading to a worsened prognosis. In this context, it would be interesting, in future research, to investigate symptom stability over time and how prognosis might also change with this."

"Commoner" sounds a little odd in American English.

This has been replaced by "more common".

We would like to highlight that as a result of these changes our manuscript now stands at 5 tables and 1 figure for a total of 4090 words. We understand that this is a little above the recommended limit, however we sincerely believe that this serves to clarify the legibility of our manuscript and hope that this is acceptable.

Thank you again for the opportunity to revise this manuscript. We look forward to hearing from you and would be happy to clarify or further review any points that you believe may benefit from further modifications.

VERSION 2 – REVIEW

REVIEWER	Tang, Sunny X Northwell Health
REVIEW RETURNED	21-Mar-2022

GENERAL COMMENTS	Thank you for addressing my comments thoroughly. One minor critique which does not preclude publication: step #4 of the flowchart indicates creation of a gold standard and a test set. What happens to the gold standard set? It seems to get lost. The subsequent steps all involve the test set.
---

REVIEWER	Levis, Maxwell White River Junction VA Medical Center
REVIEW RETURNED	31-Mar-2022

GENERAL COMMENTS	Thanks for the opportunity to re-review this compelling manuscript. The changes have improved the paper considerably, but I do think that some issues remain to be addressed. The results, discussion, and conclusion are strong, but the introduction and methods could be improved. My primary enduring concerns are addressed as follows:  1. The introduction remains quite confusing. While the other sections feel relatively cohesive, I have a hard time following the argument. Some specific notes: Could you explain FRS more directly? Can you clarify how Schneider's definition impacted the field. It's not clear what holistic and phenomenology mean in these contexts. I would recommended providing more information about clusters, symptoms, and interactions. Still not obvious who "the same authors" are. I would recommend dividing this paragraph into clearer and smaller units. I would also recommend adding background about how NLP might help address these concerns. 2. I still have concerns about lack of information about data formats, CRIS, Texthunter, annotations, and greater detail about NLP model development. It would be helpful if some details were presented in the methods and explained in the results sections. What data is being analyzed? What terms are being detected and how are they selected to reference the desired constructs? It
---

	would again be useful to clarify what type of mental health data that will be utilized. I could imagine that methods for evaluating clinical notes, outcome measure data, and pathology reports, for example, would all be different. Sadly, I am not familiar with CRIS, but it sounds like an amazing resource. The Figure is useful, but would be helpful to know more about algorithm development. What is included? Although the authors do address annotation methods and interrater reliability, it would be useful to understand how models were developed and keywords associated with the discrete constructs were identified. 3. The Venn diagrams are useful, but supporting literature and explanation in text about model conceptualization would aid argument. Where does this overlapping model come from?
--	--

VERSION 2 – AUTHOR RESPONSE

Reviewer 1:

1. Thank you for addressing my comments thoroughly. One minor critique which does not preclude publication: step #4 of the flowchart indicates creation of a gold standard and a test set. What happens to the gold standard set? It seems to get lost. The subsequent steps all involve the test set.

We have updated our methods section to clarify this by adding a new paragraph in the "Exposure variables and software development" subsection (page 11).

Reviewer 2:

1. The introduction remains quite confusing. While the other sections feel relatively cohesive, I have a hard time following the argument. Some specific notes: Could you explain FRS more directly? Can you clarify how Schneider's definition impacted the field. It's not clear what holistic and phenomenology mean in these contexts. I would recommended providing more information about clusters, symptoms, and interactions. Still not obvious who "the same authors" are. I would recommend dividing this paragraph into clearer and smaller units. I would also recommend adding background about how NLP might help address these concerns.

Thank you for this recommendation. We have rewritten our introduction to make its structure clearer, and taken this opportunity to expand on the definitions of first-rank symptoms and the importance of Schneider's definition, clarify the "same authors" formulation and further discuss the use of NLP in this context (pages 5-8).

2. I still have concerns about lack of information about data formats, CRIS, Texthunter, annotations, and greater detail about NLP model development. It would be helpful if some details were presented in the methods and explained in the results sections. What data is being analyzed? What terms are being detected and how are they selected to reference the desired constructs? It would again be useful to clarify what type of mental health data that will be utilized. I could imagine that methods for evaluating clinical notes, outcome measure data, and pathology reports, for example, would all be different. Sadly, I am not familiar with CRIS, but it sounds like an amazing resource. The Figure is useful, but would be helpful to know more about algorithm development. What is included? Although the authors do address annotation methods and interrater reliability, it would be useful to understand how models were developed and keywords associated with the discrete constructs were identified.

To explain this further, we have added a paragraph at the end of our introduction clarifying the functioning of CRIS and NLP model development as well as the type of data utilised (pages 7-8), in addition to better introducing the Text Hunter platform (page 10) and the further explanations surrounding NLP already mentioned in answer to the comments made by reviewer 1 (methods section, subsection "Exposure variables and software development" (page 11)).

3. The Venn diagrams are useful, but supporting literature and explanation in text about model conceptualization would aid argument. Where does this overlapping model come from?

The majority of studies of schizophrenia focus on a range of symptoms that commonly occur together, despite rarely describing symptom interactions directly. Our descriptive results and past evidence showing how commonly symptoms co-occur led us to analyse these interactions further and reflect this overlap in the Venn diagrams mentioned. We have added further explanations for this concept in our introduction (page 6) and our results (pages 14 and 16):

"Overall, these studies highlight the significance of positive symptoms individually and as a group, suggesting possible correlations and reflecting the overlap in their occurrence, although the degree of overlap is rarely quantified." (page 6)

"Descriptive results indicated that the symptoms of interest overlapped. This was further analysed by describing the co-occurrence of symptoms and later analysing the correlation between this and negative outcomes." (page 14).

"As mentioned previously, based on past descriptions of symptoms as co-occurring we created Venn diagrams describing their overlap. Similarly, we were interested in investigating whether our outcomes of interest occurred independently or together. Supplementary Figure 3 describes overlap between outcomes." (page 16)

We also added a sentence in our discussion where the Venn diagrams are mentioned to reflect the reasoning behind this overlapping model (page 25):

"As shown in supplementary figures 1 and 2, these symptoms overlapped in many patients as well as with other psychotic symptoms, and so did the negative outcomes under scrutiny. The overlap observed before further analyses suggested that there may exist interactions linking these factors and directed our analyses towards the development of a model to understand these better. The algorithms we developed to create this model achieved high performance metrics and our model-fit measures demonstrate model reliability but highlight that the variance remains in great part unexplained, likely due to the role of other factors not included here. This could be usefully explored in further studies to refine models assessing patient prognosis. Furthermore, our patient sample was drawn from an inner-urban South-London population and results may be different in other patient groups, although the large sample size should mitigate this effect."

Finally, we also further clarified this in the figure legends of our supplementary material (supplementary material pages 5-7):

"Supplementary Figure 1: Venn diagram describing the overlap between thought broadcasting, thought insertion and thought withdrawal. The overlap observed before further analyses suggested that there may exist interactions linking these factors and directed our analyses towards the development of a model to understand these better."

"Supplementary figure 2: Venn diagram describing the overlap between somatic passivity, thought interference and any other positive symptoms (auditory hallucinations, persecutory delusions, paranoia) . I: Thought Interference; P: Passivity; O: Other Positive Symptom. When carrying out analyses, other positive symptoms were not combined but each included as individual covariates. The overlap observed here suggested that there may exist interactions linking these factors and directed our analyses towards the development of a model to understand these better. this diagram also reflects the prevalence of other positive symptoms and the need for investigations of the possible correlation between these and outcomes in the future."

"Supplementary Figure 3: Venn diagram describing the overlap between Outcomes investigated. MHA: presence of a Mental Health act section. This diagram highlights the degree of overlap between negative outcomes, suggesting a possible correlation in their occurrence."

VERSION 3 – REVIEW

REVIEWER	Levis, Maxwell White River Junction VA Medical Center
REVIEW RETURNED	20-Jun-2022

GENERAL COMMENTS	Manuscript is much improved. Some issues remain: • Italics within table are not super helpful.• Venn diagrams are not totally clear in regard to readability and conceptual clarity. Added in-text information is not very useful in explaining the purpose of these diagrams.• Usage of P and R (from supplemental tables) should be explained more thoroughly.
--

VERSION 3 – AUTHOR RESPONSE

Thank you for the opportunity to further revise our paper, *Investigating the relationship between thought interference, somatic passivity and outcomes in patients with psychosis: a natural-language processing approach using the Clinical Record Interactive Search (CRIS) platform*. Following your recommendations, we have made the following changes:

- **Italics within table are not super helpful.**

We are happy for the tables to be altered according to the house style for the journal and for any italics to be removed, if that is preferred.

- **Venn diagrams are not totally clear in regard to readability and conceptual clarity. Added in-text information is not very useful in explaining the purpose of these diagrams.**

We are sorry that the Venn diagrams are insufficiently clear and have replaced these with tables, providing numerical values for the different overlapping/non-overlapping groups.

- **Usage of P and R (from supplemental tables) should be explained more thoroughly.**

We have added explanations and definitions for the P and R statistics in Supplementary Table 1.

Please note that with these latest changes our word count now stands at 4941. This remains a little above the recommended limit, however we sincerely believe that it serves to clarify the legibility of our manuscript and hope that this is acceptable.

Thank you again for the opportunity to revise this manuscript. We look forward to hearing from you and would be happy to clarify or further review any points that you believe may benefit from further modifications.